# Metabolite Profiles of the Relationship between Body Mass Index (BMI) Milestones and Metabolic Risk during Early Adolescence

**DOI:** 10.3390/metabo10080316

**Published:** 2020-07-31

**Authors:** Wei Perng, Mohammad L. Rahman, Izzuddin M. Aris, Gregory Michelotti, Joanne E. Sordillo, Jorge E. Chavarro, Emily Oken, Marie-France Hivert

**Affiliations:** 1Department of Epidemiology, Colorado School of Public Health, University of Colorado Denver Anschutz Medical Campus, Aurora, CO 80045, USA; 2Lifecourse Epidemiology of Adiposity and Diabetes (LEAD) Center, University of Colorado Denver Anschutz Medical Campus, Aurora, CO 80045, USA; 3Division of Chronic Disease Research Across the Lifecourse (CoRAL), Department of Population Medicine, Harvard Medical School/Harvard Pilgrim Health Care Institute, Boston, MA 02215, USA; mlr782@mail.harvard.edu (M.L.R.); Izzuddin_Aris@harvardpilgrim.org (I.M.A.); rejoa@channing.harvard.edu (J.E.S.); emily_oken@harvardpilgrim.org (E.O.); MHIVERT@partners.org (M.-F.H.); 4Metabolon Inc., Durham, NC 27560, USA; gmichelotti@metabolon.com; 5Department of Nutrition, T. H. Chan Harvard School of Public Health, Boston, MA 02115, USA; jchavarr@hsph.harvard.edu; 6Diabetes Unit, Massachusetts General Hospital, Boston, MA 02114, USA

**Keywords:** metabolites, BMI milestones, growth, metabolic risk, adolescents

## Abstract

Early growth is associated with future metabolic risk; however, little is known of the underlying biological pathways. In this prospective study of 249 boys and 227 girls, we sought to identify sex-specific metabolite profiles that mark the relationship between age and magnitude of the infancy body mass index (BMI) peak, and the childhood BMI rebound with a metabolic syndrome z-score (MetS z-score) during early adolescence (median age 12.8 years). Thirteen consensus metabolite networks were generated between male and female adolescents using weighted correlation network analysis. In girls, none of the networks were related to BMI milestones after false discovery rate (FDR) correction at 5%. In boys, age and/or magnitude of BMI at rebound were associated with three metabolite eigenvector (ME) networks comprising androgen hormones (ME7), lysophospholipids (ME8), and diacylglycerols (ME11) after FDR correction. These networks were also associated with MetS z-score in boys after accounting for age and race/ethnicity: ME7 (1.43 [95% CI: 0.52, 2.34] units higher MetS z-score per 1 unit of ME7), ME8 (−1.01 [95% CI: −1.96, −0.07]), and ME11 (2.88 [95% CI: 2.06, 3.70]). These findings suggest that alterations in sex steroid hormone and lipid metabolism are involved in the relationship of early growth with future metabolic risk in males.

## 1. Introduction

Early growth is a bellwether for future cardiovascular and metabolic health. For instance, rapid weight gain during the first two to three years of life is linked to future obesity [1,2,3,4,5,6,7], hypertension [8,9,10,11], and biomarkers of insulin-glucose homeostasis [12]. Beyond absolute change in anthropometry, milestones of growth are now recognized as indicators of future health. Early life body mass index (BMI) follows a distinct age-related pattern in most individuals, which is characterized by a rapid increase after birth, followed by attainment of the infancy BMI peak between 7 and 9 months, then falling to a nadir around 5 to 7 years of age, marking the beginning of the BMI rebound, before increasing again through late childhood/early adolescence [13,14]. Later age at the infancy BMI peak [15,16,17,18,19,20,21], earlier age at BMI rebound [21,22,23], and higher magnitude of BMI at both milestones [15,21,24] are associated with future adiposity and/or an unfavorable metabolic profile. In some cases, associations of these BMI milestones with metabolic biomarkers were independent of current BMI [20,25,26], suggesting that early growth may be related to long-term health through pathways that are independent of body composition.

A few hypotheses have been proposed to explain these associations. Some researchers posit that the age range during which growth milestones occur represent sensitive developmental periods wherein a premature rise or fall in BMI may predispose a child for obesity and obesity-related metabolic conditions later in life [27]. On the other hand, growth may simply be the manifestation of upstream risk factors, such as maternal pre-pregnancy obesity, excess gestational weight gain [28], and glucose tolerance in pregnancy [29]. A better understanding of the mechanisms or shared physiological drivers of the relationship between early growth and metabolic risk may pave the way for future studies to identify specific mechanisms to target for preventive intervention.

Here, we used data from the Project Viva observational cohort to carry out three objectives, each conducted separately for boys and girls given that the age range at outcome assessment (11–16 years) is a time when sex differences in metabolism emerge [30]. First, we confirmed the relationships of age and magnitude of the infancy BMI at peak and childhood BMI rebound (both of which are of interest based on findings of their relevance to metabolic risk in this [21] and other cohorts) with a metabolic syndrome z-score (MetS z-score) during early adolescence. For our second objective, we implemented weighted correlation network analysis (WGCNA)—a correlation-based multivariate technique that has utility for deriving meaningful latent variables from high-dimensional data—on untargeted metabolomics data to characterize consensus metabolite profiles associated with BMI milestones. Finally, of the metabolite profiles of interest (i.e., those related to the BMI milestones), our third objective was to hone in on those that are also associated with MetS z-score during early adolescence. Given the existing literature on the relevance of amino acid, acylcarnitine, lipid, and androgen hormone pathways to early growth and development [31], as well as obesity-related conditions in children and adolescents [32,33,34,35,36,37,38], we hypothesized that we would identify metabolites involved in some or all of these pathways in the present study. Key terms and concepts pertinent to this study are listed in Table 1.

We focused on the BMI peak and rebound as milestones of interest based on prior evidence from this [21] and other [15,39,40] cohorts demonstrating associations of these individual-level growth trajectory features with future cardiometabolic risk.

## 2. Results

The average age (mean ± SD) of the 476 participants was 12.9 ± 0.7 years. Approximately half (47.9%) of the sample was female, and the majority (64.0%) were of White race/ethnicity. The infancy BMI peak occurred at 8.0 ± 1.8 months for boys and 8.7 ± 3.0 months for girls. Average BMI at peak was 18.4 ± 1.4 kg/m^2^ for boys and 17.8 ± 1.3 kg/m^2^ for girls. The age at BMI rebound occurred at 5.2 ± 1.8 years for boys and 4.8 ± 1.7 years for girls. The average BMI at rebound was 16.0 ± 1.2 kg/m^2^ for boys and 15.8 ± 1.3 kg/m^2^ for girls. Additional characteristics of the study sample are displayed in Table 2.

Table 3 shows associations of BMI milestones with MetS z-score at the early teen visit. In unadjusted and multivariable models, later age at BMI peak corresponded with a higher MetS z-score in boys (β = 0.05 [95% CI: 0.00, 0.11] for both unadjusted and adjusted), and a higher magnitude of BMI at peak was associated with a greater MetS z-score in girls (unadjusted: β = 0.09 [95% CI: 0.03, 0.41]; adjusted: β = 0.08 [0.02, 0.14]). In both sexes, earlier age and higher magnitude of BMI at rebound were each associated with higher MetS z-score (Table 3). For example: in boys, each 1 SD earlier age at rebound (per 21.36 months) corresponded with a MetS z-score that was 0.20 (95% CI: 0.15, 0.25) units higher, and each 1 SD higher BMI at rebound (per 1.21 kg/m^2^) was associated with a MetS z-score that was 0.12 (95% CI: 0.07, 0.16) units higher. These MetS-related milestones were of interest in subsequent analyses.

The WGCNA analysis yielded 13 consensus networks that represent distinct metabolite networks that are relevant in both boys and girls. In Figure 1, each branch represents a hierarchical clustering of metabolites, and the colors underneath the branches correspond with the membership of a given metabolite to a network. In the subsequent text in the Results section, we refer to the modules as metabolite eigenvector (ME) 1 to 13 and interpret key metabolite networks in the Discussion.

Table 4 shows sex-specific associations of a 1 standard deviation (SD) increment in BMI milestones that were associated with MetS z-score in Table 3, which included age at the infancy BMI peak in boys and of magnitude of BMI at peak in girls. The estimates were generally null, with the exception of a positive association of BMI at peak with ME4 among girls (β = 0.009 [95% CI: 0.001, 0.017]), although this estimate was not significant after false discovery rate (FDR) correction at a 5% error rate.

Table 5 shows associations of age and magnitude of BMI at rebound (per 1 SD of each) with the network scores, separately for boys and girls. Among boys, we observed inverse associations of age at rebound with ME7 (β = −0.011 [95% CI: −0.018, −0.004]) and ME11 (β = −0.010 [95% CI: −0.018, −0.003]), and positive associations with ME8 (β = 0.014 [95% CI: 0.008, 0.021]), ME9 (β = 0.008 [95% CI: 0.001, 0.016]), and ME13 (β = 0.008 [95% CI: 0.000, 0.015]). In boys, we also found an inverse relationship of magnitude of BMI at rebound with ME5 (β = −0.010 [95% CI: −0.017, −0.002]) and ME8 (β = −0.011 [95% CI: −0.018, −0.004]). After FDR correction, age at rebound in relation to ME7, ME8, and ME11 remained significant, as did magnitude of rebound in relation to ME7. In girls, later age at rebound was associated with a higher score for ME5 (β = 0.010 [95% CI: 0.001, 0.018]) and ME8 (β = 0.010 [95% CI: 0.002, 0.018]), and a lower score for ME7 (β = −0.013 [95% CI: −0.021, −0.005]; Table 5). However, none of the estimates for girls passed FDR correction.

Table 6 shows associations of metabolite networks that were FDR significant in Table 5 with MetS z-score. In boys, higher scores for ME7 and ME11 were associated with higher MetS z-score in unadjusted and multivariable models. A 1 unit increment in the factor score for ME7 and ME11 corresponded with 1.28 (95% CI: 0.39, 2.17) and 2.83 (95% CI: 2.01, 3.65) units higher MetS z-scores, respectively, in unadjusted analyses. Accounting for age and race/ethnicity in Model 1 yielded estimates of 1.43 (95% CI: 0.52, 2.34) for ME7 and 2.88 (95% CI: 2.06, 3.70) for ME11. Additional adjustment for pubertal status and perinatal characteristics in Models 2 and 3 did not materially change the direction, magnitude, or precision of the estimates. We also noted an inverse relationship between ME8 and MetS z-score (Table 6), although the stability of this estimate was not as consistent across models as ME7 and ME11.

Table 7 shows associations of ME7, ME8, and ME11 with components of the MetS z-score in boys. Generally, the associations of the networks with individual components were consistent with those of the overall z-score. In boys, waist circumference was the component that was most consistently associated with the networks, followed by homeostatic model assessment of insulin resistance (HOMA-IR), suggesting that these two metrics may be particularly sensitive markers of metabolic risk among adolescent males.

Appendix A shows information sub- and super-pathways of the metabolites that were associated with at least one of the BMI milestones. This table also includes the network membership score for each metabolite (representative of the relative importance of a metabolite to a given network), separately for boys and girls. We include metabolites with the top 10 highest network scores for each module, although we considered all compounds with network scores >|0.6| when interpreting the networks. All top metabolites had positive scores, indicating that higher concentrations of these metabolites correspond with a higher network score for that particular network. The ranking of scores for individual metabolites within each network was similar for boys and girls, indicating that these top metabolites are equally important within a given network for both sexes.

## 3. Discussion

### 3.1. Summary of Main Findings

In this prospective study of 476 participants in the observational Project Viva cohort, we examined sex-specific associations of BMI milestones with an externally standardized metabolic syndrome score (MetS z-score) during early adolescence (median age 12.8 years) and identified metabolite networks that were associated with both early growth and metabolic risk. In both sexes, older age at BMI peak during infancy, and younger age and higher magnitude of BMI rebound during childhood, were associated with higher MetS z-score during early adolescence. These results are consistent with a previous study in a larger sample of this cohort [21], as well as those from other diverse study populations despite differences in methods for defining growth hallmarks and minor discrepancies in methods of deriving the metabolic risk endpoint. For instance, in an analysis of 163 Mexican adolescents, Perng et al. found that a later age and larger magnitude of the infancy BMI peak, derived from a Bayesian stochastic process model, were independent predictors of a metabolic syndrome risk score calculated as the average of fasting C-peptide (a surrogate for fasting insulin [42]), waist circumference, fasting glucose, the ratio of triglycerides to high-density lipoprotein (HDL), and the average of systolic (SBP) and diastolic (DBP) blood pressure [15]. In 910 Chilean children, González et al. [40] identified an early BMI rebound (<5 years) as a predictor of higher central adiposity, worse glycemia, higher triglycerides, and an adverse metabolic risk score calculated as the average of waist circumference, glucose, insulin, triglycerides, and HDL. Similarly, in a sex-specific analysis of 296 Japanese children, Koyama et al. reported that an early age at BMI rebound (<4 years), defined as the age at which the lowest BMI occurred among 15 serial BMI measurements taken between 1 and 12 years of age, was associated with an adverse lipoprotein phenotype implicated in future risk of metabolic syndrome. Specifically, the investigators found that an early BMI rebound corresponded with higher triglycerides, apolipoprotein B, and atherogenic index, and lower HDL in boys; and higher apolipoprotein B in girls [39].

When we examined sex-specific associations of each BMI milestone with 13 consensus metabolite networks generated from a weighted correlation network analysis (WGCNA), only three networks—all in boys—passed false discovery rate (FDR) correction. These networks comprised metabolites on androgen hormone (ME7), lysophospholipids (ME8), and diacylglycerol (ME11) metabolism pathways. In addition to having been previously identified as correlates of obesity sub-phenotypes in this cohort [32], individual compounds and/or pathways in the lysophospholipid and diacylglycerol patterns have been linked to cardiometabolic risk in other studies. For instance, some metabolites in the lysophospholipid pattern have been implicated in perturbed hepatic fat oxidation in young adults [43], and the diacylglycerol pattern captured several metabolites on the glycerophosphocholine pathway, which has been implicated in the development of insulin resistance and metabolic syndrome in non-human primates [44]. These metabolite networks (ME7, ME8, ME11) were also associated with MetS z-score in boys, suggesting that these pathways may be mechanisms, markers, or shared upstream physiological drivers of the relationship between early BMI growth and metabolic risk during adolescence in males.

In the following sections, we discuss the biological relevance of these three metabolite networks (ME7, ME8, and ME11) that were FDR significant and also associated with MetS z-score in boys as primary findings, followed by a discussion of secondary findings comprising other results that were nominally significant at alpha = 0.05.

### 3.2. Primary Findings (in Boys Only)

#### 3.2.1. ME7: Androgen Hormones (Associated with Age at BMI Rebound and MetS z-Score)

In boys, an earlier age at rebound, which has been related to adverse metabolic outcomes later in life [45], was associated with higher levels of compounds captured in ME7. This network comprised several androgen hormones including dehydroisoanderosterone sulfate (DHEA-S), androstenediol disulfate, and pregnenediol disulfate. We previously identified a similar metabolite profile during mid-childhood [34] and early adolescence [32] as a correlate of obesity and metabolic risk, which is consistent with findings in the present study, as this metabolite network was positively associated with MetS z-score. The steroid hormone composition of this pattern suggests that it likely represents increased androgen synthesis and a more rapid tempo of sexual maturation, which has implications for metabolic health [46].

When we examined the associations of this network with components of the MetS z-score, we noted consistency in the direction of associations across each biomarker, but with the strongest and most significant associations with waist circumference and HOMA-IR, which aligns with metabolic changes that occur during puberty, the most notable being the transient increase in insulin resistance [47,48]. The extent to which this metabolite pattern is a correlate, driver, or consequence of pubertal progression remains yet to be explored.

#### 3.2.2. ME8: Lysophospholipids (Associated with Age and Magnitude of BMI at Rebound, and MetS z-Score)

Earlier age at rebound and higher magnitude of BMI at rebound was associated with a lower score for ME8, which was composed largely of lysophospholipids, in boys. In turn, this metabolite network was inversely related to MetS z-score. Said another way, adverse hallmarks of growth (i.e., earlier age and higher BMI at rebound) were associated with lower concentrations of metabolites. In turn, lower levels of these metabolites are associated with higher metabolic risk.

Key metabolites in this network are glycerophosphocholines (i.e., 1-oleoyl-GPC (18:1), 1-steroyl-GPC (18:0), 1-linoleoyl-GPC (18:2), 1-palimotoyl-GPC (16:0)) formed in the breakdown of phosphatidylcholine and involved in basic metabolic processes such as lipid peroxidation, phospholipid metabolism, lipid transport, and fatty acid metabolism. A controlled feeding trial in eight healthy male young adults found that the consumption of α-glycerophosphocholine resulted in increased hepatic fat oxidation [43], suggesting that this particular family of metabolites may promote and enhance lipid metabolism.

Consistent with the inverse relationship between this metabolite network and the MetS z-score, ME8 was inversely related to waist circumference and positively associated with HDL. However, we found an unexpected positive association of this network with serum triglycerides, which, despite the fact that blood samples were collected in the fasting state, could reflect extraneous variability from dietary intake [49]. This association requires further investigation in a mechanistic study capable of evaluating metabolic flux (i.e., upregulation versus downregulation of specific pathways).

#### 3.2.3. ME11: Diacylglycerols (Associated with Age at BMI Rebound and MetS z-Score)

Earlier age at BMI rebound associated with a higher score for ME11, which was composed entirely of diacylglycerols (DAGs) and resembles a metabolite profile previously identified as a correlate of overweight/obesity status in conjunction with high metabolic risk in this cohort [32]. This network was associated with higher metabolic risk, suggesting that ME8 is a marker of adverse metabolic processes, potentially reflecting diet given that the top metabolites in this network are of the palmitoyl–linoleoyl–glycerol moiety, which are common emulsifiers used in bakery products, shortening, whipped toppings, and other confections [50]. Other metabolites of interest include isomers of oleoyl–linoleoyl–glycerol involved in glycerolipid metabolism [51].

Endogenously, alterations in DAG composition could be indicative of either lipolytic or lipogenic activity—a distinction that we are not able to make due to the fact that the metabolites were assessed at the same time as the metabolic risk factors. However, in a prospective study of Rhesus macaques, Polewski et al. [44] identified differences in plasma DAG composition (i.e., decreases in products of palmitate desaturation; increases in certain essential N-6 fatty acids) that served as markers of worsening insulin resistance and metabolic syndrome onset.

Indeed, assessment of this network with individual components of the risk score revealed that it was positively related to waist circumference, HOMA-IR, and triglycerides, and inversely related to HDL. We also noted the largest magnitude of association with triglycerides (i.e., β = 6.06 SD units for triglycerides per 1 unit increment in ME8, as compared to 1.8 SD units for waist circumference, 2.2 SD units for HOMA-IR, and −3.7 SD units for HDL), which makes sense given that DAGs are precursors to triglycerides. While it is tempting to speculate that this metabolite pattern represents potential disturbances in lipid metabolism, our study design hampers our ability to determine whether the DAGs in ME8 represents lipids going toward the liver for metabolism (and thus may simply be a marker of dietary intake or lifestyle) versus away from the liver (a representation of hepatic fat oxidation and lipid metabolism).

### 3.3. Secondary Findings in Boys

#### 3.3.1. ME5: Energetics (Associated with Magnitude of BMI at Rebound)

In boys, higher magnitude of BMI at rebound was related to a lower score for ME5, which captured metabolites on acylcarnitine, glutathione, and fructose/mannose/galactose metabolism pathways. The top metabolite in this network was dihomo-linolenoylcarnitine (C20:3n3 or 6), an elongation product of gamma-linolenic acid (GLA) [52]. GLA is an omega-6 polyunsaturated fatty acid (PUFA) that has been associated with slower adiposity gain in school-age children [53]. Other top loading compounds in this network include 5-oxoproline, an amino acid derived from glutathione that plays a role in glutamate storage [54]; linoleoylcarnitine (C18:2), a long-chain acyl fatty acid derivative of carnitine that is involved in the transport of long-chain fatty acids into mitochondria [55]; oleoylcarnitine (C18:1), a long-chain acylcarnitine that accumulates during certain metabolic conditions including fasting [56]; and arachidonoylcarnitine (C20:4), an acylcarnitine involved in energy cycling and storage [57]. In addition to the above metabolites that were positively correlated with the overall ME5 network score, we noted that three compounds had negative scores: glucose (−0.61), mannose (−0.62), and arginine (−0.65) (*nota bene*, these values were not shown in Appendix A as they were not among the top 10 compounds with the highest factor loadings). Such negative scores indicate that a higher score for this network corresponds with lower levels of these metabolites. Given that glucose and mannose are carbohydrates that are substrates for metabolic processes, higher levels of the top-loading compounds indicative of increased fatty acid and glutathione metabolism, in conjunction with lower levels of these carbohydrates, suggest that this network is indicative of upregulated energy turnover. While this network was inversely related to magnitude of BMI at rebound, it was not associated with concurrent metabolic risk, suggesting that the pathways captured by ME5 are not involved in the relationship between early growth and metabolic health.

#### 3.3.2. ME9: Microbial and Aromatic Amino Acid Metabolism (Associated with Age at BMI Rebound)

Earlier age at rebound was associated with a lower score for ME9 in boys. This network of metabolites represented several compounds generated by the human intestines and those involved in the metabolism of aromatic amino acids. The top metabolites in this network included phenylacetylglutamine, the primary metabolite of the degradation of phenylacetate (a catabolite of phenylalanine) when in the presence of glutamine in the liver; p-cresol glucuronide, a compound generated as an end product of tyrosine biotransformation by intestinal bacteria [58]; phenylacetylcarnitine, a substrate involved in fatty acid transport into the mitochondria; phenylacetate, an aromatic fatty acid metabolite of phenylalanine that is implicated in the activation of peroxisome proliferation-activated receptors (PPARs) and depletion of glutamine [59]; and 3-indoxyl sulfate, a metabolite synthesized from tryptophan in the human intestine [60].

While this network was not associated with metabolic risk at the same time of assessment, its composition is relevant to a growing body of literature implicating the role of the gut microbiota in moderating or mediating the relationship between serum metabolites and metabolic risk [61].

#### 3.3.3. ME13: Bile Acid Metabolism (Associated with Age at BMI Rebound)

Earlier age at rebound correlates with lower levels of metabolites in ME13, which captures compounds on primary and secondary bile metabolism pathways. The top metabolites included glycochenodeoxycholate, which is related toglycochenodeoxycholate-3-sulfate, a component of bile acid that induces oxidative stress [62]; glycocholate, a secondary bile acid produced by microbial flora in the colon [63]; and taurocholate, a sodium salt that is the chief ingredient of the bile of carnivorous animals [64]. A small case-control study in adults reported that the serum levels of these metabolites are higher in adults with non-alcoholic fatty liver disease (NAFLD; *n* = 35) than sex-matched controls (*n* = 25) [65], suggesting the involvement of bile acid metabolic pathways in the pathogenesis of NAFLD. Given that an earlier age at BMI rebound was associated with a lower score for this pattern, higher levels of these compounds may be beneficial, although we are not able to make any conclusions regarding the metabolic implications of this metabolite pattern given that ME13 was not associated with MetS z-score.

### 3.4. Secondary Findings in Girls

#### 3.4.1. ME4: Cellular Stress Response (Associated with Infancy BMI Peak)

In girls, a higher magnitude of BMI at peak was associated with a higher score for ME4, although this pattern was not associated with metabolic risk. The top metabolite in this pattern was taurine, an amino acid that is involved in one-carbon metabolism and DNA methylation pathways. Other noteworthy compounds in this network included those involved in phospholipid metabolism pathways (glycerophosphoethanolamine, phosphoethanolamine, and choline phosphate), lysoplasmalogens that are markers of synthesis and turnover of the phospholipid backbone in cell walls (1-(1-enyl-palmitoyl)-GPE (P-16:0), 1-(1-enyl-stearoyl)-GPE (P-18:0)), and inosine 5′-monophosphate (IMP), which is involved in the synthesis and degradation nucleic acids. This metabolite network resembles a metabolite pattern that we previously identified in cord blood [31], which is representative of DNA/RNA turnover and cell proliferation pathways that are upregulated during periods of cellular stress, as well as rapid growth and development.

#### 3.4.2. ME5: Energetics, ME7: Androgen Hormones, and ME8: Lysophospholipids (Associated with Age at BMI Rebound)

In girls, earlier age at BMI rebound was associated with lower levels of metabolites in ME5 (energetics) and ME8 (lysophospholipids), and higher levels of compounds in ME7 (androgen hormones). We noted that the magnitude of associations for ME7 and ME8 are similar to those detected in boys. The biological relevance of these networks is discussed in earlier sections.

None of the associations of BMI milestones with metabolite networks passed FDR correction in girls, and in turn, none of the networks were associated with MetS z-score. There are a few explanations for the null findings in girls. First, given that a large proportion of girls were already post-pubertal based on their Tanner staging (i.e., 83.6% of girls versus 45.2% of boys), it is possible that the females in this study sample have less inter-individual variability in metabolism, thereby making it more difficult to detect significant associations of the metabolite networks with MetS z-score. Additionally, even if females and males have similar total fat mass, the distribution of fat is different. Notably, females have less metabolically active central fat than males, which may further reduce variability in MetS z-score [66].

### 3.5. Strengths and Limitations

One key strength of this study was that the relatively large sample size (especially in comparison to other studies of metabolomics in youth where *N* < 300 [33,34,35,36,37,38]) enabled us to conduct sex-specific analyses, which is important given that previous findings identified sex differences in the evolution of metabolic risk biomarkers from late childhood to early adolescence in this cohort [30], and there is an established literature indicating that puberty is a time of sex-specific differences in multiple aspects of metabolic risk, including fat distribution [67,68], glucose–insulin homeostasis [48], and lipid profile [69]. Additional strengths include the prospective design with respect to the association between early growth and the metabolite networks, which provided temporal separation to avoid reverse causation bias; the implementation of a data-driven multivariate technique (WGCNA) to generate networks of metabolites assayed on an untargeted platform, which is an ideal approach for discovery of novel features related a biological trait [70]; use of the “meeting-in-the-middle” analytical strategy to identify metabolite profiles of interest (i.e., those that mark the relationship of early growth with future metabolic risk), which was recently shown to reveal novel high-dimensional biomarkers that may link exposures to disease in cohort studies [71]; the multi-ethnic study population, which may enhance the generalizability; and our ability to account for key covariates, such as pubertal status, that contribute to variability in metabolism. It was also reassuring that some metabolite networks identified herein (e.g., the androgen hormone pattern [ME7]; the cellular stress response and turnover pattern [ME4]) have now been identified at multiple life stages and using different dimension-reduction techniques in Project Viva, thereby providing the internal validity of our metabolomics analyses.

Limitations include the fact that we assessed metabolomics from fasting serum samples at the same time as the conventional risk factors used to derive the MetS z-score, which precludes our ability to untangle temporality and infer on metabolic flux. This limitation also precludes our ability to ascertain whether the metabolites network are part of the pathways leading to metabolic syndrome, a simple reflection of the differences in metabolism, or downstream consequences of subclinical metabolic syndrome. Additionally, while temporal separation between the infancy and early childhood BMI milestones and the metabolomic biomarkers reduced the possibility of reverse causation (i.e., that aberrances in metabolism may alter early growth), the long latency makes it challenging to attribute differences in metabolism during early adolescence to variation in early growth. Finally, there were some differences in the background characteristics of the study sample and the original cohort (i.e., slightly older maternal age at the index pregnancy, fewer women who smoked during pregnancy, higher annual income, and more highly educated mothers). These differences reflect sociodemographic characteristics that facilitate participation in long-term research studies (and thus may limit generalizability), but they likely do not impose concerning selection bias given that inclusion in this analysis is unlikely to be affected by the exposures (early growth, metabolite networks) or outcomes (MetS z-score) of interest in this analysis.

Areas for future inquiry include the following: (1) longer term follow-up to assess whether the metabolite networks are true mediators to the relationship between early growth and future metabolic risk; (2) untangling the extent to which the relationship between early growth and future metabolic health is independent of known upstream determinants of early growth and metabolic risk; (3) given that many metabolites of interest were lipids and/or involved in fatty acid metabolism pathways, implementing lipidomics profiling or the targeted profiling of specific lipid classes will provide a more precise quantitative assessment of the biological relevance of these pathways as they relate to early growth and metabolic risk; (4) mechanistic studies to determine whether early growth patterns are causally related to future physiology [72] or are merely markers of ongoing physiological processes; and (5) derivation and assessment of growth milestones based on trajectories of alternate anthropometric indicators such as fat mass index [73] or waist-to-height ratio [74] may shed light on the specific relationship between early life adiposity change and metabolic risk, as well as relevant metabolic pathways.

## 4. Methods

### 4.1. Study Population

This study includes participants of Project Viva, which is an ongoing observational pre-birth cohort study recruited from a multi-specialty group practice in eastern Massachusetts (Atrius Harvard Vanguard Medical Associates) [75]. There were 2128 children enrolled at birth, 1038 of whom attended the early teen research visit at age 11–16 years (median 12.8 years). Five hundred and sixty (560) of these participants had adequate volume of fasting blood for untargeted metabolomics analysis. For the present analysis, we excluded 3 participants who were considered outliers during the construction of the metabolomics networks, 54 participants missing data on BMI milestones, and 32 participants missing information on the outcome of interest (metabolic syndrome z-score). Thus, the final analytic sample comprised 476 participants.

The study sample was similar to the original 2128 enrolled participants [75] in terms of perinatal and background characteristics, with the exception of slightly older maternal age (32.7 ± 5.6 versus 31.8 ± 5.2 years), a lower proportion of mothers who smoked during pregnancy (10.1% versus 12.6%), a higher proportion of families with an annual household income >$70,000 (63.4% versus 61.2%), and a higher proportion of women who were college graduates (70.0% versus 61.2%).

All mothers provided written informed consent and children provided verbal assent. The Institutional Review Board (IRB) of Harvard Pilgrim Health Care approved all study protocols. Project Viva is registered as an observational cohort study at clinicaltrials.gov as NCT02820402.

### 4.2. Early Growth Milestones: Infancy BMI Peak and Childhood BMI Rebound

The early life growth milestones of interest are magnitude and timing of BMI peak in infancy, and of rebound in childhood. As previously described [45], we derived these metrics from children with at least three BMI measurements from birth through mid-childhood (median [range]: 7.7 [6.6–10.9] years). We calculated BMI as weight [kg]/(height [m])^2^. We measured the participants’ weights using an electronic scale to the nearest 0.1 kg (Tanita, Arlington Heights, IL) and measured standing height with a calibrated stadiometer to the nearest 1 mm (Shorr Productions, Olney, MD).

To derive the early growth milestones, we used mixed-effects models with natural cubic splines to fit individual BMI curves. We included interactions of child sex with spline terms as fixed parameters into the model to obtain sex-specific trajectories. We estimated the BMI peak in infancy and rebound in childhood as the highest and lowest points of the BMI curves where the slope equals 0. The magnitude and timing of each milestone is equivalent to the BMI (in kg/m^2^) and age (in months) of the child for each milestone. In the statistical analyses, we assessed each metric as continuous variables.

### 4.3. Blood Collection for Metabolomics Assays and Conventional Biomarkers

At the early teen visit, trained phlebotomists collected an 8-hour fasting blood sample from the antecubital vein. All samples were refrigerated immediately, processed within 24 h, and stored at −80 °C until time of analysis.

### 4.4. Untargeted Metabolomics Profiling

We carried out untargeted metabolomics profiling in fasting plasma collected at the early teen research visit via Metabolon’s multi-platform technique comprising ultra-performance liquid chromatography paired with mass spectrometry (UPLC-MS/MS) platform that interfaced with a heated electrospray ionization source and mass analyzer operated at 35,000 mass resolution [76,77,78]. Subsequently, metabolites were identified by automated comparison of the ion features in the experimental samples to a reference library of chemical standard entries that included retention time, molecular weight (*m*/*z*), preferred adducts, and in-source fragments as well as associated mass spectrometry spectra and curated by visual inspection for quality control using software developed at Metabolon. Additional details have been published [76,77,78] and are provided in Appendix A.

The laboratory analysis yielded 1135 metabolites. In the present study, we considered only endogenous metabolites, of which there were 1005 in this dataset. Prior to formal statistical analysis, we imputed missing values for metabolites as ½ the minimum detected value as we have previously done for other metabolomics datasets in Project Viva [31,32], and log_10_-transformed and Pareto-scaled each compound to remove the effect of potential outliers, as previously described [32].

### 4.5. Metabolic Risk during Early Adolescence

The outcome of interest was a metabolic syndrome z-score (MetS z-score) derived from data collected at the early teen visit. The score comprised five externally standardized biomarkers that are established risk factors for cardiometabolic disease risk: waist circumference, a measure of pernicious central adiposity that correlates well with directly-measured abdominal fat mass [79] and is an independent predictor of later life morbidity [80] and mortality [81]; the homeostatic model assessment of insulin resistance (HOMA-IR), an accurate and non-invasive measure of insulin resistance [82] that is a predictor of type 2 diabetes [83] and independent risk factor for cardiovascular disease and all-cause mortality [84] in adults; serum high-density lipoprotein (HDL) and triglycerides, both of which are cardiovascular disease risk factors that track from childhood into adulthood [85]; and systolic blood pressure, which is reliably measured in children and a predictor of future cardiometabolic health outcomes [86]. We focused on the average of these components as the outcome for both statistical and biological reasons. Statistically, using a single outcome reduces the number comparisons being made—a non-trivial task given the high-dimensional nature of metabolomics data. Biologically, beyond the relevance of each individual metric discussed above, the average of the biomarkers is associated with incident type 2 diabetes and cardiovascular mortality [87].

We measured waist circumference at the level of the umbilicus via a non-stretchable measuring tape and standardized these values using data from the CDC reference [88]. We calculated HOMA-IR (glucose *mg/dL* × insulin *µIU/mL*)/405) using fasting glucose values assessed enzymatically and fasting insulin measured via an electrochemiluminescence immunoassay (Roche Diagnostics, Indianapolis, IN). We standardized HOMA-IR values using data from 12 to 19-year-old participants of the National Health and Nutrition Examination Survey (NHANES) 1999–2002 [89]. Lipid profile was measured enzymatically and standardized according to age- and sex-specific data for participants aged 12–19 years in NHANES III. We measured SBP in quintuplicate using biannually calibrated automated oscillometric monitors (Dinamap Pro100, Tampa, Florida). We used the average of the five measurements for the statistical analysis and standardized the values using the American Academy of Pediatrics’ age-, sex-, and height-specific data [90]. After deriving the external z-scores for each of the components, we took the average across the five variables (with HDL z-score multiplied by −1) to obtain the MetS z-score.

### 4.6. Covariates and Study Population Characteristics

#### 4.6.1. Child Characteristics at the Early Teen Visit

We calculated the participants’ age based on date of the research visit and birth date. We ascertained information from the mothers on the children’s race/ethnicity via a questionnaire at the early teen research visit. Participants’ parents reported on their pubarchal/pubertal phenotype based on appearance of body hair, acne, menarche, occurrence of the pubertal growth spurt, and breast development for girls; and body hair, facial hair, acne, occurrence of the pubertal growth spurt, and deepening of voice for boys on a scale of 1 (no development) to 4 (full development). We took the average of each of these characteristics for boys and girls, separately, and entered the variable as an ordinal covariate in multivariable models. We dichotomized pubertal status as pre-pubertal (puberty score ≤ 2) versus pubertal (puberty score > 2).

#### 4.6.2. Perinatal, Sociodemographic, and Background Characteristics

At enrollment, mothers provided information on their age, pre-pregnancy weight, date of last menstrual period, education level, parity, annual household income, and smoking habits via interviews and questionnaires. We calculated pre-pregnancy BMI (kg/m^2^) using self-reported pre-pregnancy weight and height, and categorized BMI according to standard classifications of underweight (<18.5 kg/m^2^), normal weight (18.5–24.9 kg/m^2^), overweight (25.0–29.9 kg/m^2^), or obese (≥30.0 kg/m^2^) [91]. Prenatal medical records provided information on perinatal characteristics including mother’s prenatal glucose tolerance status [29], child′s sex, birthweight, and delivery date. We calculated gestational age at birth from the date of the last menstrual period or from a second trimester ultrasound if the estimated delivery date differed by >10 days. We derived birthweight-for-gestational age and sex z-scores using U.S. national reference data [41].

### 4.7. Statistical Analysis

The overarching goal of this analysis was to identify metabolite profiles that serve as markers of the relationship between BMI milestones and metabolic risk in early adolescence using a meet-in-the-middle analytical strategy [71]. First, we identified metabolite networks associated with BMI milestones. Next, we further honed in on those networks that were also associated with metabolic risk. In this section, we describe procedures for constructing metabolite networks, followed by the regression analysis used to identify metabolite networks of interest.

#### 4.7.1. Metabolite Network Analysis

Prior to regression analysis, we reduced the dimensionality of the metabolomics data using a weighted correlation network analysis (WGCNA version 1.69 in R) that has proven utility for deriving meaningful biological networks from high-dimensional data [92]. We selected this method as an unsupervised approach to construct consensus networks (a common pattern of metabolites between male and female adolescents) that account for the majority of variability in the original metabolomics dataset (and thus represent important metabolic pathways in both boys and girls), while allowing the relative importance of each metabolite network to vary between the two sexes.

All 1005 endogenous metabolites from the untargeted metabolomics dataset were entered into the network derivation algorithm based on signed Spearman correlations using a power of 10 as a weight function. The resulting network adjacency matrix [93] was transformed to a topological overlap matrix (TOM), which provides a measure of connectedness between each pair of metabolites in a network conditional on all other metabolites within that network [94]. Next, we performed hierarchical clustering on TOM-based dissimilarity and applied the Dynamic Tree Cut function [95] with a minimum module size of 20. We constructed the consensus networks modules using scale-free topology (i.e., where the structure of a network is independent of its size) in light of evidence that metabolic networks in most biological systems follow this phenomenon [96].

Upon creation of the consensus networks, we derived a score for each network using the first eigenvector (the first right-singular vector of the standardized expression profile for each network [97]) such that a higher score for a network indicates greater similarity of an individual’s metabolite profile to that represented by a given network. Since the eigenvectors are normally distributed z-scores, we used these as continuous variables representing each network in subsequent analyses.

#### 4.7.2. Regression Analysis

We ran models for conventional regression analysis separately for boys and girls given previous findings in this cohort of sex-specific changes in metabolic risk biomarkers from late childhood to early adolescence [30], and an established literature indicating that puberty is a time during which sex-specific differences in components of the MetS z-score (e.g., fat deposition [67,68], insulin resistance [48], and lipid profile [69]) become apparent.

#### 4.7.3. Step 1: Identify Growth Milestones Associated with MetS z-Score

First, we examined the association of the four BMI milestones (age and magnitude of peak BMI in infancy, and of the BMI rebound in childhood) with MetS z-score at the early teen visit. We examined unadjusted associations followed by multivariable adjustment for race/ethnicity and age at the early teen visit, and we were interested in associations that were statistically significant at alpha = 0.05 given the prior knowledge regarding each of these milestones.

#### 4.7.4. Step 2: Identify Metabolite Networks Associated with Growth Milestones

Next, among the BMI milestones that were associated with MetS z-score, we identified metabolite networks that were associated with the milestones using linear regression models where each BMI milestone was the independent variable and each metabolite network was the dependent variable. In these models, we accounted for the child’s race/ethnicity and age at the early teen visit. We noted metabolite networks that were associated with BMI milestones at alpha = 0.05 but focused our interpretation of results on networks that were significant after false discovery rate (FDR) correction at 5%.

#### 4.7.5. Step 3: Identify Growth-Related Metabolite Networks Associated with MetS z-Score

Among the networks that were associated with BMI milestones, we examined associations of these networks with MetS z-score using unadjusted and multivariable linear regression models. While numerous lifestyle (e.g., diet, physical activity, sleep) and environmental (e.g., exposure to toxicants, air pollution) characteristics have the potential to impact the relationships among early growth and metabolism, the goal of this analysis is to identify metabolite profiles that mark this relationship. Thus, we focused on a parsimonious set of biological covariates to avoid “adjusting away” meaningful biological variation. In Model 1, we adjusted for age at early teen visit and the child’s race/ethnicity. In Model 2, we further accounted for the child’s pubertal status at the early teen visit. In Model 3, we included covariates in Model 2 plus perinatal characteristics known to be associated with offspring adiposity and metabolism: maternal education level (college graduate yes versus no), pre-pregnancy BMI (continuous), and smoking habits (former smoker, never smoker, or smoked during pregnancy).

If we found that a metabolite network was significantly associated with MetS z-score, we also examined its associations with individual components of the score. For this analysis, we adjusted for covariates in the model and did not correct for multiple comparisons given that each of the components are correlated aspects of the same construct.

#### 4.7.6. Sensitivity Analyses

In sensitivity analyses, we mutually adjusted the contemporaneous BMI milestones for one another (i.e., adjusting the estimate for age of a given milestone for magnitude of BMI at the milestone and vice versa) in models where the growth milestones were the exposure and the metabolite networks were the outcomes. These analyses did not yield markedly different direction, magnitude, or precision of the estimates, so we did not include these variables in the final models.

All analyses were performed using the Statistical Analyses System 9.4 software (SAS Institute Inc., Cary, NC, USA) unless otherwise indicated.

## 5. Conclusions

In this analysis of BMI milestones, metabolite profiles, and metabolic risk, we identified three metabolite networks representing androgen hormones, lysophospholipids, and DAGs that may represent important biological pathways involved in the relationship between early growth and metabolic risk in boys.

## Figures and Tables

**Figure 1 metabolites-10-00316-f001:**
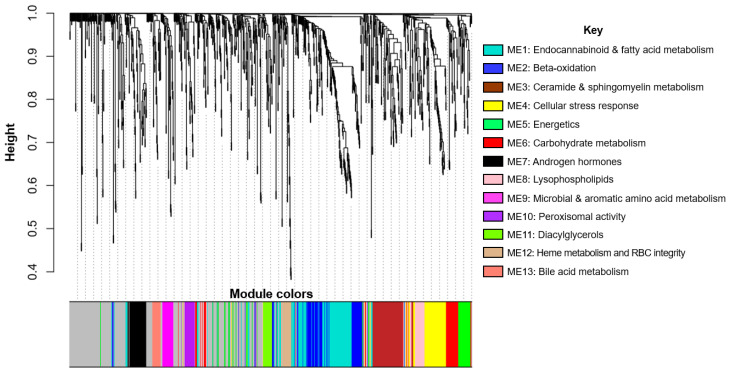
**Dendrogram of consensus metabolite networks generated by weighted gene co-expression network analysis (WGCNA).** Composition of key metabolite networks (i.e., those associated with early BMI milestones and/or MetS z-score) are in Appendix A.

**Table 1 metabolites-10-00316-t001:** Key terms and concepts.

Term/concept	Explanation
Infancy body mass index (BMI) peak	An individual-level BMI trajectory milestone characterized by a vertex that typically occurs between 7 and 9 months of age. Later age and higher magnitude of this milestone has been associated with future obesity and metabolic risk.
Childhood BMI rebound	An individual-level BMI trajectory milestone characterized by a nadir that typically occurs between 3 and 6 years of age. Earlier age and higher magnitude of this milestone has been related to future obesity and metabolic risk.
False discovery rate (FDR)	A statistical procedure that protects against type 1 error (i.e., false positive findings) when making multiple comparisons. The correction requires an investigator-defined threshold of acceptable type 1 error. In this study, we used an error rate of 5%.
Untargeted metabolomics platform	An unbiased laboratory approach to measuring low-molecular-weight compounds (metabolites) in a biosample. This type of platform typically involves mass spectrometry (MS) paired with liquid (LC) and/or gas chromatography (GC) to estimate relative concentrations of all detectable metabolites (from hundreds to thousands) in a biosample. The broad coverage of this type of platform makes it ideal for hypothesis generation and discovery of new biomarkers.
Metabolic syndrome (MetS) z-score	A metabolic syndrome risk z-score calculated as the average of externally age- and sex-standardized waist circumference, systolic blood pressure, reversed high-density lipoprotein (HDL), the homeostatic model assessment of insulin resistance (HOMA-IR), and triglycerides.
Weighted correlation network analysis (WGCNA)	A correlation-based dimension reduction procedure that constructs latent variables (“consensus networks”) that account for the majority of variability in a high-dimensional dataset.
Meeting-in-the-middle	An analytical strategy that involves identifying a features (e.g., metabolites) from a high-dimensional dataset (e.g., untargeted metabolomics data) that are associated with an exposure, and also associated with an outcome.

**Table 2 metabolites-10-00316-t002:** Background characteristics of 476 Project Viva participants.

	Mean ± SD or % (N)	
	Boys	Girls
	*n =* 249	*n* = 227
***Perinatal and Family Characteristics***		
Mother′s age at enrollment (years)	32.7 ± 5.6	32.7 ± 4.9
Annual household income > $70,000	63.9% (145)	62.8% (130)
Mother is college graduate	69.0% (161)	71.2% (161)
Maternal pre-pregnancy BMI (kg/m^2^)	24.8 ± 5.2	24.9 ± 5.3
Gestational diabetes mellitus diagnosis	4.4% (11)	4.0% (9)
Mother smoked during pregnancy	11.2% (28)	8.9% (20)
Birthweight-for-gestational age z-score ^a^	0.26 ± 1.01	0.18 ± 0.97
***Growth hallmarks***		
Age at infancy BMI peak (months)	8.0 ± 1.8	8.7 ± 3.0
BMI at peak (kg/m^2^)	18.4 ± 1.4	17.8 ± 1.3
Age at BMI rebound (years)	5.2 ± 1.8	4.8 ± 1.7
BMI at rebound (kg/m^2^)	16.0 ± 1.2	15.8 ± 1.3
***Child characteristics at the early teen visit***		
Age (years)	12.9 ± 0.6	13.0 ± 0.7
Race/ethnicity		
Black	16.1% (40)	12.8% (29)
Hispanic	4.4% (11)	4.0% (9)
White	61.8% (154)	66.4% (150)
Asian	2.8% (7)	3.1% (7)
>1 race/ethnicity	14.9% (37)	13.7% (31)
Puberty score >2	45.2% (112)	83.6% (189)
Waist circumference (cm)	73.0 ± 11.2	72.6 ± 10.4
SBP (mmHg)	108.7 ± 8.9	105.7 ± 8.8
HDL (mg/dL)	56.2 ± 13.7	55.9 ± 12.5
HOMA-IR	2.75 ± 1.73	3.42 ± 2.23
Triglycerides (mg/dL)	66.8 ± 30.9	71.7 ± 29.8
MetS z-score^b^	−0.13 ± 0.43	−0.20 ± 0.43

Abbreviations: BMI: body mass index, HDL: high-density lipoprotein; HOMA-IR: homeostatic model assessment of insulin resistance; MetS z-score: metabolic syndrome z-score; SBP: systolic blood pressure. a—According to Oken et al. [41]. b—Externally standardized metabolic syndrome z-score comprised of waist circumference, systolic blood pressure, reversed HDL, triglycerides, and HOMA-IR.

**Table 3 metabolites-10-00316-t003:** Unadjusted and adjusted associations of growth milestones (per 1 SD) with metabolic syndrome z-score at the early teen visit (median age 12.8 years) among boys and girls in Project Viva ^a^.

	Unadjusted				Adjusted ^b^			
	Boys (*n* = 249)		Girls (*n* = 227)		Boys (*n* = 248)		Girls (*n* = 225)	
	β (95% CI)	*P*	β (95% CI)	*P*	β (95% CI)	*P*	β (95% CI)	*P*
**BMI peak**								
Age	**0.05 (0.00, 0.11)**	**0.05**	−0.02 (−0.07, 0.04)	0.51	**0.05 (0.00, 0.11)**	**0.05**	−0.03 (−0.08, 0.03)	0.36
Magnitude	0.01 (−0.04, 0.07)	0.59	**0.09 (0.03, 0.14)**	**0.001**	0.01 (−0.04, 0.07)	0.64	**0.08 (0.02, 0.14)**	**0.005**
**BMI rebound**								
Age	**−0.19 (−0.24, −0.14)**	**<0.0001**	**−0.10 (−0.16, −0.05)**	**0.0002**	**−0.20 (−0.25, −0.15)**	**<0.0001**	**−0.09 (−0.15, −0.04)**	**0.0008**
Magnitude	**0.12 (0.07, 0.17)**	**<0.0001**	**0.12 (0.07, 0.17)**	**<0.0001**	**0.12 (0.07, 0.17)**	**<0.0001**	**0.11 (0.06, 0.17)**	**<0.0001**

^a^—Boys: 1 SD age at infancy BMI peak = 1.85 mo, 1 SD BMI at peak = 1.44 kg/m^2^, 1 SD age at BMI rebound = 21.36 mo, BMI at rebound = 1.21 kg/m^2^; Girls: 1 SD age at infancy BMI peak = 3.02 mo, 1 SD BMI at peak = 1.32 kg/m^2^, 1 SD age at BMI rebound = 20.34 mo, 1 SD BMI at rebound = 1.26 kg/m^2^; ^b^—Estimates are adjusted for race/ethnicity and age at the early teen visit. Bolded values indicate statistical significance at alpha = 0.05.

**Table 4 metabolites-10-00316-t004:** Associations of age and magnitude of BMI at peak (per 1 SD increment in each metric) with metabolite network scores derived from untargeted metabolomics data at the early teen visit (median age 12.8 years) for boys and girls in Project Viva ^a^.

	Boys		Girls	
	*Y*-Variable: Age at Peak		*Y*-Variable: Magnitude of BMI at Peak	
*X-*variable	*β (95% CI) per 1.85 mo*	*P*	*β (95% CI) per 1.32 kg/m^2^*	*P*
ME1	−0.001 (−0.005, 0.003)	0.51	0.000 (−0.008, 0.008)	0.99
ME2	−0.0057 (−0.013, 0.002)	0.12	0.004 (−0.004, 0.012)	0.37
ME3	−0.006 (−0.013, 0.001)	0.11	−0.003 (−0.010, 0.005)	0.54
ME4	−0.002 (−0.009, 0.005)	0.57	**0.009 (0.001, 0.017)**	**0.03**
ME5	0.001 (−0.007, 0.008)	0.84	0.004 (−0.004, 0.012)	0.34
ME6	−0.004 (−0.011, 0.004)	0.23	0.003 (−0.005, 0.010)	0.52
ME7	0.003 (−0.004, 0.011)	0.36	0.003 (−0.005, 0.011)	0.52
ME8	−0.004 (−0.011, 0.004)	0.33	0.001 (−0.007, 0.009)	0.77
ME9	−0.002 (−0.009, 0.006)	0.67	−0.002 (−0.010, 0.006)	0.67
ME10	−0.003 (−0.010, 0.004)	0.43	0.003 (−0.005, 0.011)	0.49
ME11	0.005 (−0.002, 0.013)	0.17	0.001 (−0.007, 0.009)	0.88
ME12	−0.006 (−0.013, 0.001)	0.09	−0.003 (−0.011, 0.005)	0.43
ME13	0.003 (−0.004, 0.010)	0.44	0.003 (−0.006, 0.011)	0.52

^a^—Estimates are adjusted for race/ethnicity and age at the early teen visit. Bolded values indicate statistical significance at alpha = 0.05.

**Table 5 metabolites-10-00316-t005:** Associations of age and magnitude of the childhood BMI rebound (per 1 SD increment in each metric) with metabolite network scores based on untargeted metabolomics data at the early teen visit (median age 12.8 years) for boys and girls in Project Viva ^a^.

	Boys			
	*Y*-Variable: BMI Rebound			
	Age		Magnitude	
	*β (95% CI) per 21.4 mo*	*P*	*β (95% CI) per 1.21 kg/m^2^*	*P*
***X-*variable**				
ME1	−0.004 (−0.011, 0.003)	0.29	0.000 (−0.007, 0.008)	0.90
ME2	0.003 (−0.005, 0.010)	0.50	−0.003 (−0.011, 0.004)	0.37
ME3	−0.002 (−0.009, 0.006)	0.63	0.000 (−0.007, 0.007)	0.97
ME4	0.000 (−0.008, 0.007)	0.93	0.000 (−0.008, 0.007)	0.99
ME5	0.000 (−0.008, 0.008)	0.99	**−0.010 (−0.017, −0.002)**	**0.01**
ME6	0.007 (0.000, 0.014)	0.06	−0.002 (−0.009, 0.005)	0.62
ME7	**−0.011 (−0.018, −0.004) ***	**0.003 ***	0.004 (−0.004, 0.011)	0.35
ME8	**0.014 (0.008, 0.021) ***	**<0.0001 ***	**−0.011 (−0.018, −0.004) ***	**0.002 ***
ME9	**0.008 (0.001, 0.016)**	**0.03**	−0.005 (−0.013, 0.002)	0.16
ME10	0.007 (−0.001, 0.014)	0.07	−0.007 (−0.014, 0.000)	0.06
ME11	**−0.010 (−0.018, −0.003)**	**0.007 ***	0.004 (−0.003, 0.012)	0.25
ME12	0.004 (−0.004, 0.011)	0.31	−0.003 (−0.010, 0.005)	0.45
ME13	**0.008 (0.000, 0.015)**	**0.04**	−0.005 (−0.012, 0.002)	0.19
	**Girls**			
	***Y*-Variable: BMI Rebound**			
	**Age**		**Magnitude**	
	***β (95% CI) per 20.3 mo***	***P***	***β (95% CI) per 1.26 kg/m^2^***	***P***
***X*-variable**				
ME1	0.003 (−0.006, 0.011)	0.55	0.003 (−0.004, 0.011)	0.40
ME2	0.000 (−0.008, 0.009)	0.98	0.002 (−0.006, 0.010)	0.62
ME3	0.002 (−0.007, 0.010)	0.67	−0.001 (−0.009, 0.006)	0.74
ME4	0.004 (−0.004, 0.013)	0.32	0.006 (−0.003, 0.014)	0.18
ME5	**0.010 (0.001, 0.018)**	**0.03**	−0.002 (−0.010, 0.006)	0.63
ME6	−0.003 (−0.011, 0.005)	0.47	0.004 (−0.003, 0.012)	0.29
ME7	**−0.013 (−0.021, −0.005)**	**0.002**	0.007 (0.000, 0.015)	0.06
ME8	**0.010 (0.002, 0.018)**	**0.02**	−0.006 (−0.014, 0.002)	0.14
ME9	0.005 (−0.004, 0.013)	0.27	−0.005 (−0.013, 0.003)	0.22
ME10	0.000 (−0.009, 0.008)	0.93	0.004 (−0.003, 0.010)	0.27
ME11	0.003 (−0.005, 0.011)	0.48	0.000 (−0.008, 0.008)	0.92
ME12	0.005 (−0.003, 0.013)	0.25	−0.003 (−0.010, 0.005)	0.44
ME13	0.004 (−0.005, 0.012)	0.38	−0.001 (−0.009, 0.007)	0.84

^a^—Estimates are adjusted for race/ethnicity and age at the early teen visit. Bolded values denote statistical significance at alpha = 0.05.*—Denotes statistical significance after false discovery rate (FDR) correction at 5%.

**Table 6 metabolites-10-00316-t006:** Associations of key metabolite networks with a metabolic syndrome z-score (MetS z-score) among boys in Project Viva at the early teen visit (median age 12.8 years).

	Difference in MetS z-Score per 1 Unit Metabolite Network in Boys							
	Unadjusted (*n* = 249)		Model 1 (*n* = 248)		Model 2 (*n* = 248)		Model 3 (*n* = 247)	
Metabolite Network	β (95% CI)	*P*	β (95% CI)	*P*	β (95% CI)	*P*	β (95% CI)	*P*
ME7	**1.28 (0.39, 2.17)**	**0.005**	**1.43 (0.52, 2.34)**	**0.002**	**1.16 (0.19, 2.13)**	**0.02**	**1.18 (0.25, 2.12)**	**0.01**
ME8	−0.92 (−1.86, 0.02)	0.06	**−1.01 (−1.96, −0.07)**	**0.04**	**−0.95 (−1.88, −0.01)**	**0.05**	−0.56 (−1.42, 0.30)	0.20
ME11	**2.83 (2.01, 3.65)**	**<0.0001**	**2.88 (2.06, 3.70)**	**<0.0001**	**2.79 (1.98, 3.61)**	**<0.0001**	**2.79 (2.00, 3.59)**	**<0.0001**

Model 1: Adjusts for child′s age race/ethnicity and age at the early teen visit. Model 2: Model 1 + pubertal status. Model 3: Model 2 + maternal education level, pre-pregnancy BMI, and smoking habits during pregnancy. Bolded values indicate statistical significance at alpha = 0.05.

**Table 7 metabolites-10-00316-t007:** Associations of metabolite networks with metabolic syndrome z-score components among boys in Project Viva at the early teen visit (median age 12.8 years) ^a.^

	Waist Circumference z-Score		HOMA-IR z-Score		HDL z-Score		Triglyceride z-Score		SBP z-Score	
	β (95% CI)	*P*	β (95% CI)	*P*	β (95% CI)	*P*	β (95% CI)	*P*	β (95% CI)	*P*
**Metabolite network**										
ME7	**2.27 (1.02, 3.52)**	**0.0004**	**2.05 (0.25, 3.85)**	**0.03**	−1.30 (−2.77, 0.16)	0.08	0.11 (−1.15, 1.37)	0.86	1.40 (−0.35, 3.15)	0.12
ME8	**−3.76 (−4.99, −2.52)**	**<0.0001**	0.14 (−1.73, 2.01)	0.88	**2.91 (1.43, 4.38)**	**0.0001**	**1.47 (0.19, 2.75)**	**0.02**	−0.02 (−1.84, 1.79)	0.98
ME11	**1.81 (0.59, 3.02)**	**0.004**	**2.24 (0.51, 3.97)**	**0.01**	**−3.70 (−5.05, −2.35)**	**<0.0001**	**6.06 (5.11, 7.00)**	**<0.0001**	0.59 (−1.10, 2.29)	0.49

Abbreviations: HDL: high-density lipoprotein; HOMA-IR: homeostatic model assessment of insulin resistance; MetS z-score: metabolic syndrome z-score; SBP: systolic blood pressure. ^a^—Estimates are adjusted for race/ethnicity and age at the early teen visit. Bolded values indicate statistical significance at alpha = 0.05.

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
