# Peer review of "Metabolite Profiles of the Relationship between Body Mass Index (BMI) Milestones and Metabolic Risk during Early Adolescence"

_metabolites, 2020, doi:10.3390/metabo10080316_

Round 1

Reviewer 1 Report

In this manuscript, the authors analyzed the relationship between the metabolomic profile, BMI milestones and metabolic risk. The experiment is well designed. The large sample size in this study makes the conclusion very convincing. It seems that, most of the metabolites showing the strong correlation between MetS z-score are lipids. The authors may perform lipidomic profiling in the future. Overall, I would like to recommend accept this manuscript after minor revison. 

Author Response

Comment: In this manuscript, the authors analyzed the relationship between the metabolomic profile, BMI milestones and metabolic risk. The experiment is well designed. The large sample size in this study makes the conclusion very convincing. It seems that, most of the metabolites showing the strong correlation between MetS z-score are lipids. The authors may perform lipidomic profiling in the future. Overall, I would like to recommend accept this manuscript after minor revision.

Response
: We thank the reviewer for the positive feedback. Indeed, many of the metabolites within networks of interest are lipids. In response to the reviewer’s suggestion for future lipidomics analyses, we have included the following text in the Discussion section:

Page 14 lines 392-396: “…; (3) given that many metabolites of interest were lipids and/or involved in fatty acid metabolism pathways, implementing lipidomics profiling or targeted profiling of specific lipid classes will provide a more precise quantitative assessment of the biological relevance of these pathways as they relate to early growth and metabolic risk;…”

Reviewer 2 Report

Summary:

It must be completed with the changes indicated in the development of the article.

Development of the article:

In the introduction section

The introduction does not correspond to the depth of the study carried out, it must be completed to justify the need to carry out the research study that is being presented and what impact the information presented has, identifying one or several defined hypotheses. One aspect that should be clarified is the selection of metabolic biomarkers in the comparison. This will allow authors to focus the article better, synthesize the information presented in the discussion and better justify the conclusions presented.

It is recommended that the authors open a concepts section where concepts that need to be clarified are briefly explained, this information can be included in the introduction or in the methods section.

In the results section

The recommendations in the introduction may require changes to this section.

In the discussion section

It is recommended that this section be synthesized and that a narrative and comparative writing be carried out with previous research.

Approaches to future studies should be related to the information shown in the article, the rest of the information should be eliminated.

In the methodology section

In this section it has not been indicated what type of clinical trial is being discussed, other than the temporal and geographical information, although this information must be completed with the previous article already published (Pre-, Perinatal, and Parental Predictors of Body Mass Index Trajectory Milestones. J Pediatr. 2018).

Details that catch my attention:

  1. Neither this article nor the previous one indicates when the information that is being presented has been registered (Pre-, Perinatal, and Parental Predictors of Body Mass Index Trajectory Milestones. J Pediatr. 2018).
  2. It is not indicated what type of clinical trial it is.
  3. The procedures for measuring weight, height and waist circumference are not described (Body composition and morphological assessment of nutritional status in adults: a review of anthropometric variables. J Hum Nutr Diet 2016, 29 (1), 7 ‐25. DOI: 10.1111 / jhn.12278).

In the conclusions section

Here a general conclusion should be made, the results, specific conclusions and direction for future research should be presented in the discussion, as indicated in the authors' instructions.

Author Response

Reviewer #2

  1. Comment: The introduction does not correspond to the depth of the study carried out, it must be completed to justify the need to carry out the research study that is being presented and what impact the information presented has, identifying one or several defined hypotheses. One aspect that should be clarified is the selection of metabolic biomarkers in the comparison. This will allow authors to focus the article better, synthesize the information presented in the discussion and better justify the conclusions presented.

  2. Response: We have modified the last paragraph of the Introduction to better integrate concepts (including a reference to a concepts section, Box 1) and included a hypothesis. To avoid making the Introduction too lengthy, we added text in the Methods to provide rationale for the biomarkers selected as part of the metabolic syndrome z-score.

    Introduction:

Pages 2 lines 64-79:  Here, we used data from the Project Viva observational cohort to carry out three objectives, each conducted separately for boys and girls given that the age range at outcome assessment (11-16 years) is a time when sex differences in metabolism emerge (30). First, we confirmed the relationships of age and magnitude of the infancy BMI at peak and childhood BMI rebound (both of which are of interest based on findings of their relevance to metabolic risk in this (21) and other cohorts) with a metabolic syndrome z-score (MetS z-score) during early adolescence. For our second objective, we implemented weighted correlation network analysis (WGCNA) – a correlation-based multivariate technique that has utility for deriving meaningful latent variables from high dimensional data – on untargeted metabolomics data to characterize consensus metabolite profiles associated with BMI milestones. Finally, of the metabolite profiles of interest (i.e., those related to the BMI milestones), our third objective was to home in on those that are also associated with MetS z-score during early adolescence. Given existing literature on the relevance of amino acid, acylcarnitine, lipid, and androgen hormone pathways to early growth and development (31), as well as obesity-related conditions in children and adolescents (32-38), we hypothesized that we would identify metabolites involved in some or all of these pathways in the present study. Key terms and concepts pertinent to this study are listed in Box 1.

Methods:

Page 16 lines 453-468: The outcome of interest was a metabolic syndrome z-score (MetS z-score) derived from data collected at the early teen visit. The score comprised five externally-standardized biomarkers that are established risk factors for cardiometabolic disease risk: waist circumference, a measure of pernicious central adiposity that correlates well with directly-measured abdominal fat mass (78) and is an independent predictor of later life morbidity (79) and mortality (80); the homeostatic model assessment of insulin resistance (HOMA-IR), an accurate and non-invasive measure of insulin resistance (81) that is a predictor of type 2 diabetes (82) and independent risk factor for cardiovascular disease and all-cause mortality (83) in adults; serum high density lipoprotein (HDL) and triglycerides, both of which are cardiovascular disease risk factors that track from childhood into adulthood (84); and systolic blood pressure, which is reliably measured in children and a predictor of future cardiometabolic health outcomes (85). We focused on the average of these components as the outcome for both statistical and biological reasons. Statistically, using a single outcome reduces the number comparisons being made – a non-trivial task given the high-dimensional nature of metabolomics data. Biologically, beyond the relevance of each individual metric discussed above, the average of the biomarkers is associated with incident type 2 diabetes and cardiovascular mortality (86).

  1. Comment: It is recommended that the authors open a concepts section where concepts that need to be clarified are briefly explained, this information can be included in the introduction or in the methods section.

    Response: We now include Box 1, which defines key concepts and terms. We reference this addition at the end of the Introduction.

  2. Comment: The recommendations in the introduction may require changes to this section (Results section).

    Response: We have made the appropriate changes throughout the body of the manuscript following changes to the Introduction.

  3. Comment: It is recommended that this section (Discussion) be synthesized and that a narrative and comparative writing be carried out with previous research.

    Response: At the start of the Discussion, we have provided a broad and integrative overview of the primary findings of the analysis. We added new text on how our findings align with existing literature on early BMI growth and future metabolic risk (see below – underlined font denotes new text). Following this summary, we include subsections discussing specific findings with respect to each metabolite network in the context of existing biochemical and epidemiological literature to explain the biological meaning of each network in the context of early growth and development of metabolic risk.

    Page 10 lines 171-207: In this prospective study of 476 participants in the observational Project Viva cohort, we examined sex-specific associations of BMI milestones with an externally-standardized metabolic syndrome risk score (MetS z-score) during early adolescence (median age 12.8 years), and identified metabolite networks that were associated with both early growth and metabolic risk. In both sexes, older age at BMI peak during infancy, and younger age and higher magnitude of BMI rebound during childhood were associated with higher MetS z-score during early adolescence. These results are consistent with a previous study in a larger sample of this cohort (31), as well as those from other diverse study populations despite differences in methods for defining growth hallmarks and minor discrepancies in methods of deriving the metabolic risk endpoint. For instance, in an analysis of 163 Mexican adolescents, Perng et al. found that later age and larger magnitude of the infancy BMI peak, derived from a Bayesian stochastic process model, were independent predictors of a metabolic syndrome risk score calculated as the average of fasting C-peptide (a surrogate for fasting insulin (53)), waist circumference, fasting glucose, the ratio of triglycerides to HDL, and the average of SBP and DBP (15). In 910 Chilean children, González et al. (33) identified an early adiposity rebound (<5 years) as a predictor of higher central adiposity, worse glycemia, higher triglycerides, and an adverse metabolic risk score calculated as the average of waist circumference, glucose, insulin, triglycerides, and HDL. Similarly, in a sex-specific analysis of 296 Japanese children, Koyama et al. reported that an early age at adiposity rebound (<4 years), defined as the age at which lowest BMI occurred among fifteen serial BMI measurements taken between 1 and 12 years of age, was associated with an adverse lipoprotein phenotype implicated in future risk of metabolic syndrome. Specifically, the investigators found that an early adiposity rebound corresponded with higher triglycerides, apolipoprotein B, and atherogenic index ([total cholesterol – HDL]/HDL), and lower HDL in boys; and higher apolipoprotein B in girls (32).

When we examined sex-specific associations of each BMI milestone with 13 consensus metabolite networks generated from a weighted correlation network analysis (WGCNA), only three networks – all in boys – passed false discovery rate (FDR) correction. These networks comprised metabolites on androgen hormone (ME7), lysophospholipids (ME8), and diacylglycerol (ME11) metabolism pathways. In addition to having been previously identified as correlates of obesity sub-phenotypes in this cohort (46), individual compounds and/or pathways in the lysophospholipid and diacylglycerol patterns have been linked to cardiometabolic risk in other studies. For instance, some metabolites in the lysophospholipid pattern have been implicated in perturbed hepatic fat oxidation in young adults (57), and diacylglycerol pattern captured several metabolites on the glycerophosphocholine pathway, which has been implicated in development of insulin resistance and metabolic syndrome in non-human primates (61). These metabolite networks (ME7, ME8, ME11) were also associated with MetS z-score in boys, suggesting that these pathways may be mechanisms, markers, or shared upstream physiological drivers of the relationship between early BMI growth and metabolic risk during adolescence in males. 

  1. Comment: (In the Discussion) Approaches to future studies should be related to the information shown in the article, the rest of the information should be eliminated.

    Response: We have simplified our suggestions for future studies and removed superfluous information (page 14 lines 389-397).

  2. Comment: In this section (Methods) it has not been indicated what type of clinical trial is being discussed, other than the temporal and geographical information, although this information must be completed with the previous article already published (Pre-, Perinatal, and Parental Predictors of Body Mass Index Trajectory Milestones. J Pediatr. 2018).

    Response: The study population, Project Viva, is an observational pre-birth cohort study, not a clinical trial. We now mention the observational nature of this cohort throughout the manuscript.

  3. Comment: Neither this article nor the previous one indicates when the information that is being presented has been registered (Pre-, Perinatal, and Parental Predictors of Body Mass Index Trajectory Milestones. J Pediatr. 2018).

    Response: Project Viva was registered as an observational cohort study at clinicaltrials.gov as NCT02820402. We have now added this information on page 15 line 417.

  4. Comment: It is not indicated what type of clinical trial it is.

    Response: This study is not a clinical trial and we do not assign interventions to the study participants.

  5. Comment: The procedures for measuring weight, height and waist circumference are not described (Body composition and morphological assessment of nutritional status in adults: a review of anthropometric variables. J Hum Nutr Diet 2016, 29 (1), 7 ‐ DOI: 10.1111 / jhn.12278).

    Response: These details are included the Methods section:

Page 15 lines 422-424: We calculated BMI as weight [kg]/(height [m])2. We measured the participants’ weights using an electronic scale to the nearest 0.1 kg (Tanita, Arlington Heights, IL) and measured standing height with a calibrated stadiometer to the nearest 1 mm (Shorr Productions, Olney, MD).

Page 16 lines 469-470: We measured waist circumference at the level of the umbilicus via a non-stretchable measuring tape, and standardized these values using data from the CDC reference (82)

  1. Comment: Here (Conclusion section) a general conclusion should be made, the results, specific conclusions and direction for future research should be presented in the discussion, as indicated in the authors' instructions.

    Response: We have omitted text on specific conclusions and direction for future research to the Discussion section and simplified the general conclusion (page 18 lines 579-582).

Round 2

Reviewer 2 Report

Authors are encouraged to continue with this type of research, however, for future research studies it is recommended to follow the following recommendations.

Following a measurement protocol in circumferences or skinfolds due to biases may occur (Madden AM, Smith S. Body composition and morphological assessment of nutritional status in adults: a review of anthropometric variables. J Hum Nutr Diet. 2016; 29 (1 ): 7-25. Doi: 10.1111 / jhn.12278). On the other hand, it seems that the ratio of waist circumference and height is a more accurate marker for calculating body fat (Santos S, Severo M, Lopes C, Oliveira A. Anthropometric Indices Based on Waist Circumference as Measures of Adiposity in Children. Obesity (Silver Spring). 2018; 26 (5): 810–813. Doi: 10.1002 / oby.22170).

Currently, BMI is not the most appropriate anthropometric parameter to assess metabolic risk, as indicated in the following studies:

  1. Javed A, Jumean M, Murad MH, et al. Diagnostic performance of body mass index to identify obesity as defined by body adiposity in children and adolescents: a systematic review and meta-analysis. Pediatr Obes. 2015;10(3):234‐244. doi:10.1111/ijpo.242
  2. Liu P, Ma F, Lou H, Liu Y. The utility of fat mass index vs. body mass index and percentage of body fat in the screening of metabolic syndrome. BMC Public Health. 2013;13:629. doi:10.1186/1471-2458-13-629
  3. Alpízar M, Frydman TD, Reséndiz-Rojas JJ, Trejo-Rangel MA, Aldecoa-Castillo JM. Fat Mass Index (FMI) as a Trustworthy Overweight and Obesity Marker in Mexican Pediatric Population. Children (Basel). 2020;7(3):19. doi:10.3390/children7030019

From there they must select another method to obtain body fat and fat-free mass, the following research will help them: Müller MJ, Braun W, Pourhassan M, Geisler C, Bosy-Westphal A. Application of standards and models in body composition analysis. Proc Nutr Soc. 2016; 75 (2): 181-7. doi: 10.1017 / S0029665115004206.

And with the following study they will be able to more accurately analyze the body composition of the subjects to be analyzed: Müller MJ, Bosy-Westphal A. Effect of Over- and Underfeeding on Body Composition and Related Metabolic Functions in Humans. Curr Diab Rep. 2019; 19 (11): 108. doi: 10.1007 / s11892-019-1221-7

Based on the arguments presented, it should be reported that due to the limitations of the BMI and the waist circumference, this study should be repeated with the analysis of the body fat index and the waist index.

Author Response

Point-by-point response to reviewer comments:

Comment: Authors are encouraged to continue with this type of research, however, for future research studies it is recommended to follow the following recommendations.

Following a measurement protocol in circumferences or skinfolds due to biases may occur (Madden AM, Smith S. Body composition and morphological assessment of nutritional status in adults: a review of anthropometric variables. J Hum Nutr Diet. 2016; 29 (1 ): 7-25. Doi: 10.1111 / jhn.12278). On the other hand, it seems that the ratio of waist circumference and height is a more accurate marker for calculating body fat (Santos S, Severo M, Lopes C, Oliveira A. Anthropometric Indices Based on Waist Circumference as Measures of Adiposity in Children. Obesity (Silver Spring). 2018; 26 (5): 810–813. Doi: 10.1002 / oby.22170).

Currently, BMI is not the most appropriate anthropometric parameter to assess metabolic risk, as indicated in the following studies:

  1. Javed A, Jumean M, Murad MH, et al. Diagnostic performance of body mass index to identify obesity as defined by body adiposity in children and adolescents: a systematic review and meta-analysis. Pediatr Obes. 2015;10(3):234‐ doi:10.1111/ijpo.242
  2. Liu P, Ma F, Lou H, Liu Y. The utility of fat mass index vs. body mass index and percentage of body fat in the screening of metabolic syndrome. BMC Public Health. 2013;13:629. doi:10.1186/1471-2458-13-629
  3. Alpízar M, Frydman TD, Reséndiz-Rojas JJ, Trejo-Rangel MA, Aldecoa-Castillo JM. Fat Mass Index (FMI) as a Trustworthy Overweight and Obesity Marker in Mexican Pediatric Population. Children (Basel). 2020;7(3):19. doi:10.3390/children7030019

From there they must select another method to obtain body fat and fat-free mass, the following research will help them: Müller MJ, Braun W, Pourhassan M, Geisler C, Bosy-Westphal A. Application of standards and models in body composition analysis. Proc Nutr Soc. 2016; 75 (2): 181-7. doi: 10.1017 / S0029665115004206.

And with the following study they will be able to more accurately analyze the body composition of the subjects to be analyzed: Müller MJ, Bosy-Westphal A. Effect of Over- and Underfeeding on Body Composition and Related Metabolic Functions in Humans. Curr Diab Rep. 2019; 19 (11): 108. doi: 10.1007 / s11892-019-1221-7

Based on the arguments presented, it should be reported that due to the limitations of the BMI and the waist circumference, this study should be repeated with the analysis of the body fat index and the waist index.

Response: Our use of BMI measurements from birth through mid-childhood to derive the early growth milestones of interest is rooted in the fact that the trajectory of BMI growth in children is well-established (as evidenced by growth references developed by the WHO and CDC) and certain characteristics of the BMI trajectory (i.e., the infancy peak and childhood nadir referred to as the “rebound”) have been linked to future obesity and metabolic risk, making them of interest in the present analysis. For the exact reasons that the reviewer mentioned, we are careful not to refer to these milestones as metrics of adiposity. In other words, we use the term “infancy BMI peak” rather than “infancy adiposity peak,” and say “childhood BMI rebound” rather than “childhood adiposity rebound” throughout. We agree that it would be potentially interesting to assess milestones of trajectories based on other metrics of body composition that are known to align closely with adiposity. Thus, we have included the following text and the references provided by the reviewer in our suggestions for future directions:

Page 14 lines 398-401: “…(5) derivation and assessment of growth milestones based on trajectories of alternate anthropometric indicators such as fat mass index (72) or waist-to-height ratio (73) may shed light on the specific relationship between early life adiposity change and metabolic risk, and relevant metabolic pathways.